# Internal Jugular Vein Tumor Thrombus: A Tricky Question for the Thyroid Surgeon

Jean-Baptiste Morvan [1,*] , Laurys Boudin [2] , Denis Metivier [3] , David Delarbre [4] , Edouard Bouquillon [1] , Juliette Thariat [5] , Damien Pascaud [1] and Pierre-Yves Marcy [6]

1  Department of Head and Neck Surgery, University Military Hospital Sainte-Anne, 2, Boulevard Sainte Anne, BP 600, 83000 Toulon, France
2  Department of Oncology, University Military Hospital Sainte-Anne, 2, Boulevard Sainte Anne, BP 600, 83000 Toulon, France
3  Department of Nuclear Medicine, University Military Hospital Sainte-Anne, 2, Boulevard Sainte Anne, BP 600, 83000 Toulon, France
4  Department of Internal Medicine, University Military Hospital Sainte-Anne, 2, Boulevard Sainte Anne, BP 600, 83000 Toulon, France
5  Department of Radiation Oncology, Comprehensive Cancer Center Francois Baclesse, 14000 Caen, France
6  Polyclinics ELSAN Group, Department of Radiodiagnostics and Interventional Imaging, PolyClinics Les Fleurs, Quartier Quiez, 83189 Ollioules, France
*  Correspondence: jbmorvan@hotmail.com; Tel.: +33-672-823-662

**Abstract:** Internal jugular vein tumor thrombus is an extremely rare condition in thyroid carcinoma, but it does exist. Correlated with greater aggressiveness with a higher incidence of distant metastases at diagnosis and a higher recurrence rate, this important prognostic element should be systematically investigated by ultrasound operators in all patients presenting with thyroid carcinoma. The patient's follow-up must be careful. This can be a trap that surgeons must look for in their preoperative checklist. We report the case of a 58-year-old woman with an IJV thrombus associated with multiple bone metastases. She underwent successful surgical treatment, and postoperative pathology showed a poorly differentiated follicular carcinoma of the thyroid and a tumor thrombus in the internal jugular vein.

**Keywords:** thyroid cancer; venous thrombosis; internal jugular vein; ultrasound scan

## 1. Introduction

Follicular thyroid carcinoma (FTC) can be the result of microscopic vascular invasion. In papillary thyroid carcinoma (PTC), lymphatic spread is more common than hematogenous spread. Distant organ metastasis is rare, in particular in PTC [1]. The risk of intravenous tumor thrombus has been different among various malignancies. In thyroid cancer, thrombosis causes are multiple [2]: extrinsic compression [3], angioinvasion [4], or possibly a prothrombotic state with hypercoagulability [5]. However, literature on thyroid cancer and the risk of venous thromboembolism is rare. Tumor thrombus and particularly internal jugular vein (IJV) tumor thrombus is an extremely rare condition in thyroid carcinoma, but it does exist. In general, tumor thrombi are seen in FTC-based thyroid cancer, and IJV invasion is seen in PTC-based thyroid cancer [6]. However, there seem to be some reports of tumor thrombus in PTC though rare. In the 2022 Gui review [7], authors counted forty-seven cases in the English literature between 1972 and 1 May 2021.

Correlated with greater aggressiveness at diagnosis and a higher recurrence rate, it is an important prognostic element [4]. Invasion into great cervical veins from a local thyroid carcinoma recurrence is even more rare but has a high mortality rate [8]. It is a diagnostic challenge for ultrasound (US) operators in the evaluation of a thyroid mass, especially when there are signs of extrathyroidal extension. This is also a trap that surgeons must not

overlook in their preoperative checklist in order to anticipate surgical difficulties on the vascular level, in particular at the skull base and the basicervical region.

We report the case of a 58-year-old woman with an IJV thrombus associated with multiple bone metastases of thyroid origin. She underwent successful surgical treatment, and postoperative pathology showed a poorly differentiated follicular carcinoma of the thyroid and a tumor thrombus into the IJV lumen.

## 2. Case Report

A 58-year-old woman with no previous medical history consulted with chronic low back pain. Spinal imaging revealed a pathological fracture of the third lumbar vertebra on standard radiography, without dural involvement on MRI. $^{18}$F-FDG PET scanner (Figure 1) confirmed multiple osteolytic lesions of the lumbar spine, 7th right rib, and left scapula, associated with an intense hypermetabolic focus (SUV max = 16.3) in the right thyroid lobe, with a second right hypermetabolic focus compatible with an inferior jugulo-carotid adenopathy of 9 mm (SUV max = 13.2). She had no other symptoms. The patient had no family history of thyroid carcinoma and no history of radiation exposure in childhood. The physical examination showed the right nodule, which is palpable, non-fixed, and flexible. There was no clinical cervical adenopathy and no compressive syndrome. Laboratory tests showed that thyroxine and thyroid-stimulating hormone levels were within normal limits. Serum calcitonin level was under 10 ng/L.

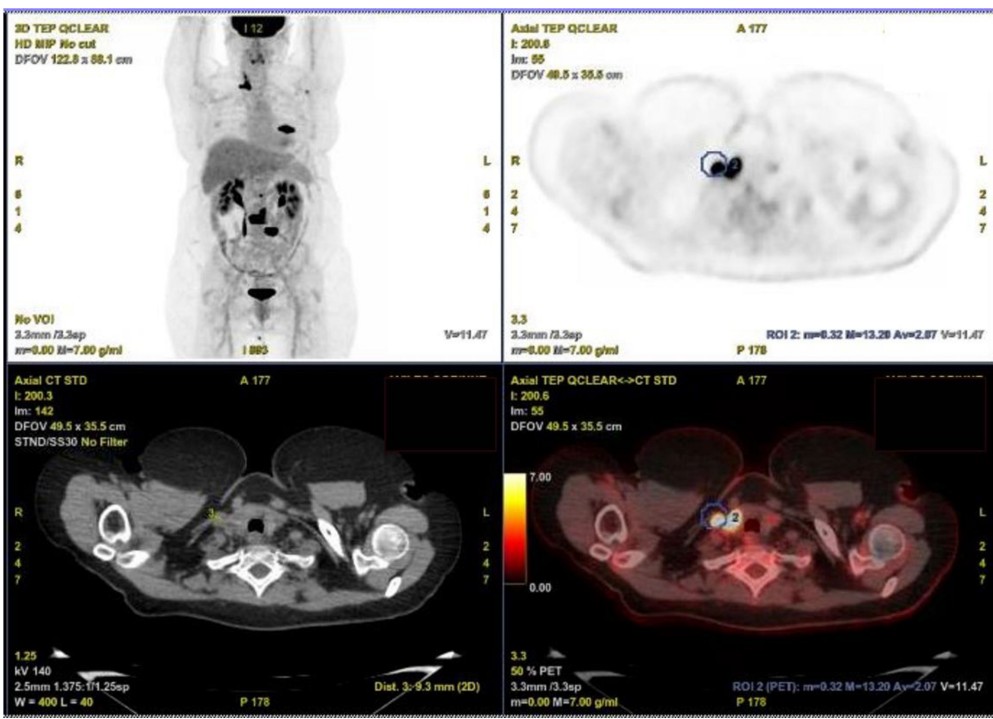

**Figure 1.** $^{18}$F-FDG PET/CT scanner. Intense hypermetabolic focus (SUV max = 16.3) of the right thyroid lobe, with a second right hypermetabolic focus compatible with a specific thrombotic tumor invasion of the right IJV (9 mm, SUV max = 13.2) in the blue circle.

Neck ultrasound confirmed a 15 mm hypoechogenic nodular formation of the right lower lobe of the thyroid, with anterior capsular contact. This nodule was classified as 5 according to the European-Thyroid Imaging Reporting and Data System EU-TIRADS published in 2017 by the European Thyroid Association. There was no suspicious adenopathy at US assessment and the second hypermetabolic focus reported on the PET/CT scanner corresponded to a specific thrombotic tumor invasion of the right IJV via the right inferior thyroid vein. There was another specific and distinct thrombotic invasion of the right IJV by the right middle thyroid vein (Figure 2a). This bifocal thrombotic invasion of the right IJV

was reported on a diagram (Figure 2b). Under the Valsalva manoeuvre, the IJV continued to dilate without signs of complete thrombotic occlusion.

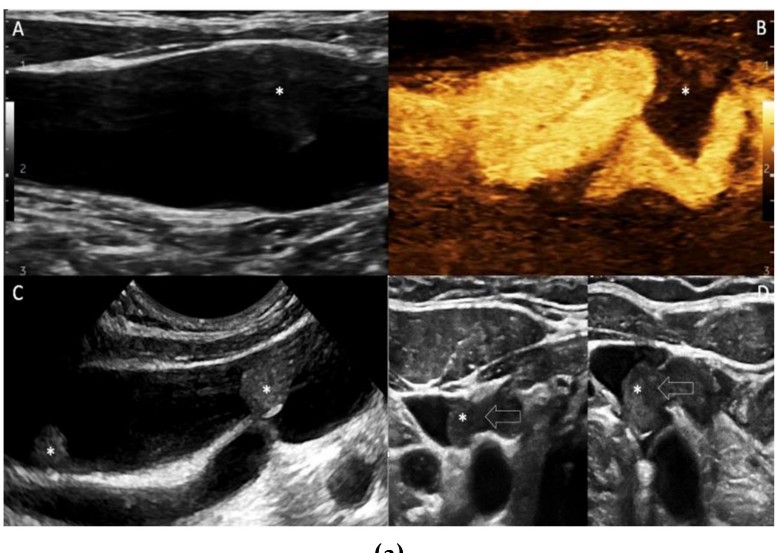

(a)

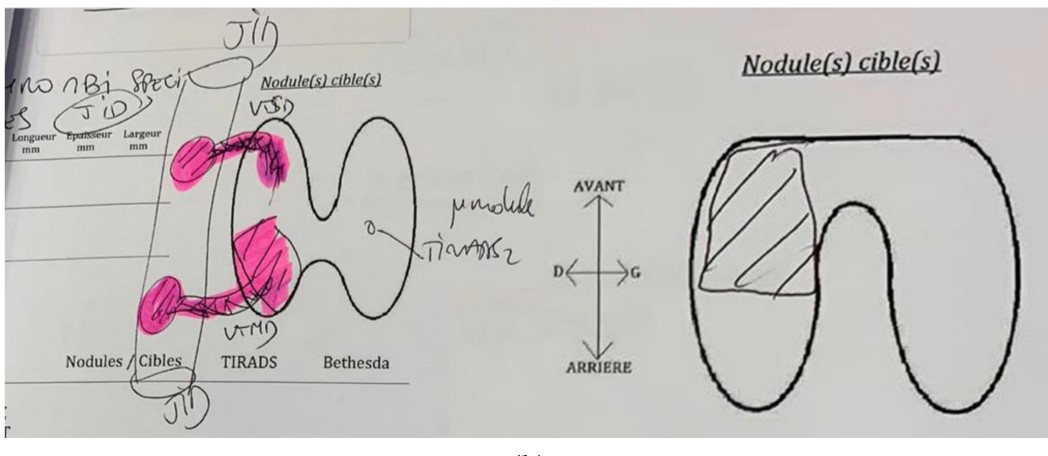

(b)

**Figure 2.** (**a**) Neck ultrasound. A: Sagittal B-mode US scanning of right IJV. Note that intra-venous thrombus (*) is poorly depictable on common settings. B: Sagittal MicroVascular Imaging (MVI) of the right IJV. MVI is a novel imaging technique (reverse color mode) that is more sensitive to depict lower velocities. Tumor thrombus (*) is shown as a black signal void into the IJV lumen (appears in white color). C: Sagittal B-mode US scanning of the right IJV is performed with a higher frequency transducer. Focusing on the IJV clearly depicts two different thrombi located at the mid and lower right thyroid. D: High frequency (16 MHz) axial B-mode US scanning displays enlarged thyroid veins at mid and lower thyroid (arrows), which empty into the ipsilateral IJV. (**b**) Neck ultrasound diagram. Thyroid nodule and bifocal thrombotic invasion of the right IJV.

Nodule thyroid ultrasound-guided Fine Needle Aspiration (FNA) cytology identified a poorly differentiated papillary carcinoma, a malignant category (VI) according to the 2017 BETHESDA system for Reporting Thyroid Cytopathology [9]. A scan-guided biopsy of the lumbar pedicle confirmed the bone metastasis of thyroid origin (poorly differentiated follicular carcinoma).

Anticoagulant treatment was started. After a multidisciplinary team meeting, the surgical management consisted of total thyroidectomy with the removal of the infiltrated portion of the IJV, in monoblock united by the thrombi infiltrating the middle and inferior right thyroid veins (Figure 3). The IJV was ligated both inferiorly and superiorly and at a distance from the thrombi and excised. A central compartment and right selective neck

dissection (levels 2–4) was performed. Intraoperative vagal continuous neuromonitoring was used. The postoperative course was simple with no complications.

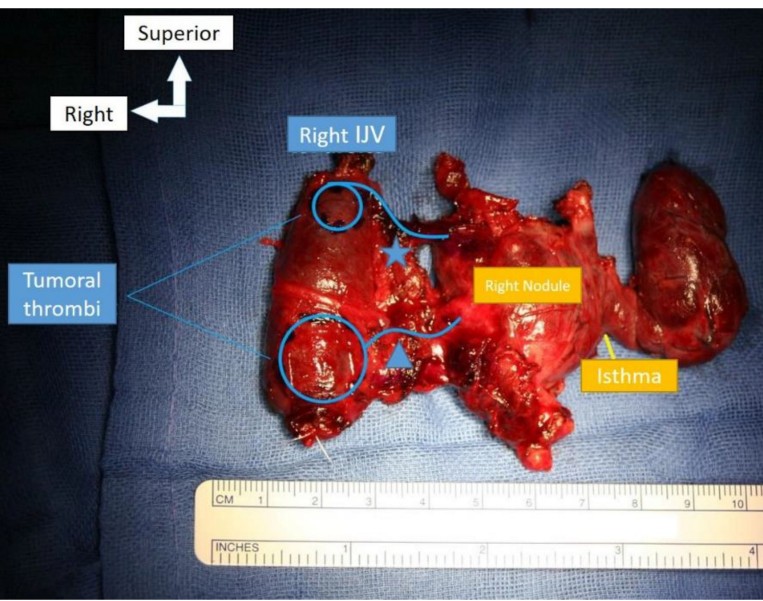

**Figure 3.** Total thyroidectomy with partial resection of the internal jugular vein (IJV) after ligature. Bifocal thrombotic invasion of the IJV by the middle (*) and inferior (Δ) thyroid veins.

Post-operative pathology (Figure 4) showed a poorly differentiated follicular thyroid carcinoma (according to the Turin criteria and 2017 World Health Organization (WHO) classification of tumors of endocrine organs [1]), largely invasive with extrathyroidal extension, measuring 20 mm. The carcinomatous proliferation was poorly differentiated in sheets, sometimes finely lobulated by capillaries, giving an insular appearance, with some follicular architecture, reworked by necrosis. The tumor appeared partially limited by a fibrous capsule, largely infiltrated and protruding, extending into the adjacent parenchyma in the form of small satellite foci. It invaded the perithyroid tissue in the inferior right and external poles, coming into contact with the muscle and invading the IJV, creating a tumor thrombus at this level. There was no lymph node involvement. The lesion was classified as pT3bN0M1 (eighth edition of the UICC TNM staging system).

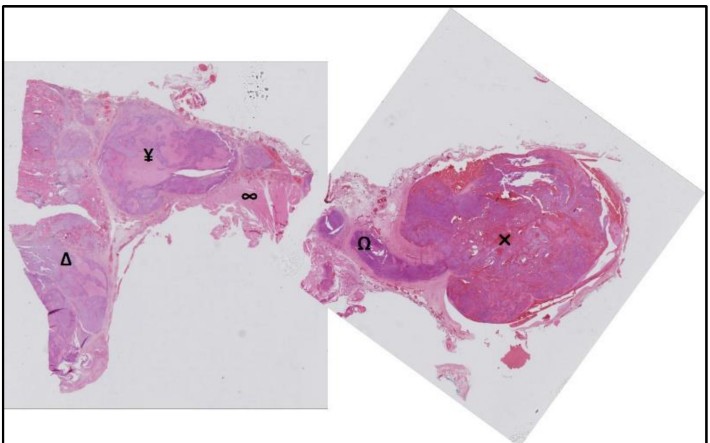

**Figure 4.** Post-operative pathology. Poorly differentiated follicular thyroid carcinoma (Δ) with extrathyroidal extension (¥) coming into contact with the muscle (∞) and invading the IJV via the middle vein thyroid (Ω), creating a tumor thrombus (×) at this level.

The patient received radioactive iodine therapy (131I) after TSH stimulation. Six months after, no tumor recurrence or distant metastasis was observed.

The authors obtained appropriate informed consent and permissions from the patient in order to include case details and images of the patient in this article.

## 3. Discussion

This case report illustrates that IJV tumor thrombus is an extremely rare condition in thyroid carcinoma that can be a pitfall that surgeons should not overlook in their preoperative checklist in order to anticipate surgical difficulties on the vascular level, in particular at the skull base and the basicervical region. It is a diagnostic challenge for US operators who should search for it in all patients with thyroid carcinoma.

Direct extraluminal vascular invasion with FTC is extremely rare and can vary in extent. In almost all cases, intravascular tumor extension usually starts at the thyroid veinous pedicles with intraluminal invasion by malignant cells and deposition of fibrin, leading to continued growth [10]. Then, the thrombus is located in the large vessels in close proximity to the thyroid and can vary in extent up to the right atrium: external jugular veins, IJV, brachiocephalic vein, innominate vein, subclavian vein, axillary vein, superior vena cava [2]. In rare occasions, tumor embolism can affect remote anatomical sites such as the cardiopulmonary area (with pulmonary vein, ascending aorta, pulmonary artery, valvular endocardium, and right ventricle) or middle cerebral artery [11]. Clinical presentation in great cervical veins can vary from asymptomatic presentation, as in our patient, to cervical and arm edema with pain, up to florid superior vena cava syndrome [8,12]. The superior vena cava occlusion clinical incidence ranged from 0.34 to 0.8%, mostly due to aggressive tumors including dedifferentiated or anaplastic thyroid carcinomas [13].

In Gui's review [7], as our patient, women predominate in 74% of cases. The median age was 62 years (from 26 years to 84 years).

In our case, the $^{18}$F-FDG PET scanner was wrongly in favour of a suspicious lateral cervical adenopathy. The rereading and comparison during the neck ultrasound examination allowed us to correct the tumoral thrombus diagnosis and depict two different levels of IJV invasion. However, the usefulness of PET and PET/CT in the detection of a tumor thrombus has been reported for various tumors [6] such as lung cancer, renal cell cancer, hepatocellular cancer, and osteosarcoma but also follicular, medullary, or anaplastic thyroid carcinoma [14–16]. Wang [16] described a case in which an $^{18}$F-FDG PET/CT scan was more sensitive for the diagnosis of a tumor thrombus despite a negative posttherapy whole-body scintigraphy with elevated unstimulated thyroglobulin 7 months after $^{131}$I ablation and thyroidectomy for PTC.

Preoperative thyroid US is a multiparametric examination and plays an important role in the diagnosis [17]. Operators should look for the presence of tumor thrombi in all patients with thyroid carcinoma. They should start by using a low-frequency common US probe and then improve the sensitivity by using higher dedicated US probes. Doppler spectral analysis is essential and provides information about the location and degree of occlusion of the central vein lumen by assessing spectral waveforms of subclavian veins and IJV on both sides [10]. "Direct intraluminal spread" is the primary tumor invasion from the medial thyroid vein into the IJV lumen without invading the IJV wall. The classic differential diagnosis is intra-thyroidal metastasis of renal origin with venous invasion [18]. Marcy [19] advocates for IJV/superior cava vena US assessment using the Valsalva maneuver to depict intraluminal grayscale thrombus and color Doppler and spectral analysis to diagnose tumor thrombus malignancy and the extent to the superior cava vena.

The most effective treatment for aggressive thyroid cancers is total thyroidectomy, possibly extended, followed by TSH-suppressive therapy and radioactive iodine (RAI) therapy with $^{131}$I ablation. There is no treatment standard for tumor thrombi in metastatic thyroid carcinoma. Intraluminal extension is not a contraindication to aggressive surgical treatment in well-differentiated thyroid cancers due to the relatively good prognosis of differentiated thyroid cancer as well as to a decreased risk of superior vena cava obstruction,

sudden death from airway occlusion, tumor embolism or fatal right atrial obstruction [20]. Surgery with vascular repair offers better survival and recovery rates rate in comparison with nonsurgical management [21]. IJV ligation is possible if it is performed on one side only and does not pose any difficulty, as in the case of our patient. IJV vein reconstruction has been advocated to avoid fatal complications according to bilateral jugular vein ligation [21].

PTC displays gross venous invasive features in 1.5% of histopathologic specimens but this vascular invasion is an important prognostic element [19]. Patients with extrathyroidal vascular invasion had a more aggressive behavior with a higher incidence of distant metastases at diagnosis, as in our patient who featured with initial multiple bone metastases. According to Kobayashi et al. [17], the patients presenting with tumor thrombi were more likely to have pulmonary metastasis. For all types of thyroid cancer, the risk of locoregional recurrence and distant metastasis is greater when there is an extrathyroidal extension, especially when there is clear evidence of invasion of surrounding structures. This is a more significant association than the presence of intrathyroidal vascular invasion. Nevertheless, patients presenting with tumors identified to have intrathyroidal vascular invasion were more likely to develop distant recurrence [22]. The reported survival of patients who display invasion of IJV or other great cervical veins by thyroid cancer ranges between 2 and 5 years [8]. Authors strongly suggest careful follow-up of these patients [7].

## 4. Conclusions

IJV tumor thrombus is an extremely rare condition in FTC, but it does exist and can be a trap for the surgeon before surgery. US operators should report on the patency of the large vessels in the neck and check for the presence of tumor thrombi in all patients with thyroid carcinoma.

It is an important prognostic element with a higher incidence of distant metastases at diagnosis. The patient's follow-up is important, given the higher rate of distant recurrence.

**Author Contributions:** Conceptualization, J.-B.M., E.B., D.P. and P.-Y.M.; validation, L.B., J.T. and P.-Y.M.; writing—original draft preparation, J.-B.M. and E.B.; writing—review and editing, J.-B.M., E.B., D.P., L.B., D.D. and D.M.; supervision, J.-B.M. and P.-Y.M. All authors have read and agreed to the published version of the manuscript.

**Funding:** This research received no external funding.

**Institutional Review Board Statement:** Not applicable.

**Informed Consent Statement:** Written informed consent has been obtained from the patient(s) to publish this paper.

**Data Availability Statement:** Not applicable.

**Conflicts of Interest:** The authors declare no conflict of interest.

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
