# Peer review of "Internal Jugular Vein Tumor Thrombus: A Tricky Question for the Thyroid Surgeon"

_curroncol, doi:10.3390/curroncol29120723_

Round 1

Reviewer 1 Report

The authors summarized the important literature in the Discussion section along with the report of a case of thyroid cancer with tumor embolization.

However, the manuscript contains major problems.

1. There are no novel content. Though tumor thrombi are indeed rare, there have been many reports on them for a long time. What is the unique/original topic that this case was focused on? The imaging diagnosis, the surgical ingenuity, the complications caused by the tumor thrombus, or the postoperative outcome?

2. Pathological figure (i.e. HE) to support histological subtype of thyroid cancer should be included. In general, tumor thrombi are seen in FTC-based thyroid cancer, and IJV invasion are seen in PTC-based thyroid cancer as mentioned in citation 13. (However, there seems to be some reports of tumor thrombus in PTC though rare.) These two should be clearly explained to avoid confusion. Pathological figures are important for these reasons.

Other minor point

The difficulty in preoperative tumor thrombus diagnosis depends on the extent of the thrombus. The diagnosis is easy when thrombus is large. It is unclear why the preoperative evaluation of a tumor thrombus is important from this report.

Also, when the M1 had already been diagnosed, active monitoring will undoubtedly be performed. (This is a comment to line23-34.)

What do you mean by the “trap” as described in line 24 and others? If the authors means that the “trap” is pitfall, what is the lessons learned from this case report? Please describe them clearly.

Is postoperative follow-up differ depending on the presence or absence of a tumor plug in M1 cases at diagnosis as described in line 212-?

Tumor size in line 50 in PET-CT imaging is unnecessary due to the US evaluated size of 15mm has been described in line 58.

What is curative anticoagulation? Tumor thrombus should be treated separately to venous thromboembolism (VTE). Accordingly, reference 5 and 8 are not appropriate for reference.

The topic about TKIs (lines 189-195) is not relevant and inappropriate to this case report.

The topics the authors intend to describe in this case report should be clarified. The contents of the discussion section are too broad and not coherent.

Author Response

The authors summarized the important literature in the Discussion section along with the report of a case of thyroid cancer with tumor embolization.

However, the manuscript contains major problems.

  1. There are no novel content. Though tumor thrombi are indeed rare, there have been many reports on them for a long time. What is the unique/original topic that this case was focused on? The imaging diagnosis, the surgical ingenuity, the complications caused by the tumor thrombus, or the postoperative outcome?

This case report was focused on imaging diagnosis.

It illustrates that internal jugular vein tumor thrombus is an extremely rare condition in thyroid carcinoma that can be a pitfall that surgeons should not overlook in their preoperative checklist in order to anticipate surgical difficulties on the vascular level. It is a diagnostic challenge for US operators who should search it in all patients with thyroid carcinoma.

  1. Pathological figure (i.e. HE) to support histological subtype of thyroid cancer should be included. In general, tumor thrombi are seen in FTC-based thyroid cancer, and IJV invasion are seen in PTC-based thyroid cancer as mentioned in citation 13. (However, there seems to be some reports of tumor thrombus in PTC though rare.) These two should be clearly explained to avoid confusion. Pathological figures are important for these reasons.

This point has been corrected in the manuscript. There is already a pathological figure (Figure 4) of the case wich is a poorly differentiated thyroid carcinoma.

Other minor point

・The difficulty in preoperative tumor thrombus diagnosis depends on the extent of the thrombus. The diagnosis is easy when thrombus is large. It is unclear why the preoperative evaluation of a tumor thrombus is important from this report.

It is important for the surgeon not to ignore the presence of a tumour thrombus whatever its size

Also, when the M1 had already been diagnosed, active monitoring will undoubtedly be performed. (This is a comment to line23-34.)

・What do you mean by the “trap” as described in line 24 and others? If the authors means that the “trap” is pitfall, what is the lessons learned from this case report? Please describe them clearly.

This point has been clearly corrected in the manuscript.

Is postoperative follow-up differ depending on the presence or absence of a tumor plug in M1 cases at diagnosis as described in line 212-?

Follow-up does not differ but it is important given higher incidence of distant metastases at diagnosis and higher rate of distant recurrence.

・Tumor size in line 50 in PET-CT imaging is unnecessary due to the US evaluated size of 15mm has been described in line 58.

this item has been removed

・What is curative anticoagulation? Corrected by “Anticoagulation treatment”

Tumor thrombus should be treated separately to venous thromboembolism (VTE). Accordingly, reference 5 and 8 are not appropriate for reference.

・The topic about TKIs (lines 189-195) is not relevant and inappropriate to this case report.

this paragraph has been removed

・The topics the authors intend to describe in this case report should be clarified. The contents of the discussion section are too broad and not coherent.

The discussion plan has been corrected

Reviewer 2 Report

This is an interesting case and an interesting topic generally. However, the logical flow of your presentation and particularly the discussion is lost because of your problems with writing in English. Personally, I have great respect for anyone writing in a language not their own and I do appreciate the difficulties. Sometimes it is the way a sentence is phrased, and sometimes it is single words (for example, polars, line 88; outcome, line 126; muti parametric, line 164).

The thrust of the case is that this is a ”tricky question” (title) or “diagnostic trap” (line 164) for the thyroid surgeon. Surely it is more a challenge for diagnosticians (radiologists and ultrasonographers mostly) as part of the assessment of any neck or thyroid lump, particularly when there is any evidence of extrathyroidal extension? It is also better not to refer to a thyroid surgeon as male (“his” in line 151) – “the preoperative checklist” would avoid this.

The Introduction just repeats some of the Abstract, it would be better not to do this but to set the scene by talking about the presence of extrathyroidal extension (more usually into oesophagus or trachea when a major structure is affected) and how common (or rare) this is, and that it only occurs in a small proportion of patients with differentiated thyroid cancer (i.e. papillary / follicular) and a higher proportion of those with medullary or anaplastic cancer. That might be the point to introduce the idea that venous thrombosis can be the direct effect of tumour invasion but might alternatively relate to a hypercoagulable state. Introducing the idea of propagation of thrombus down into the SVC or cranially would be good here so that the reader can understand the risk of unrecognised venous thrombosis. It would be good also to describe here (with any appropriate references) what the consequences of unrecognised neck vein thrombosis actually are. It is less uncommon to find IJV/EJV thrombosis in head and neck cancers as an incidental finding, and in most cases, there are no consequences. The risk of pulmonary embolism (as compared to leg vein thrombosis) seems very low.

The actual case is well described though hard to follow in places. The use of the word “bifocal” throughout the paper is excessive. I am not sure the risks are greater from tumour invasion at two sites compared to one. Figure 1 (the image of  the PET/CT scan) is too small, and there should be a legend to explain what the blue circle on the image is showing. The ultrasound images in Figure 2a are less helpful (except to radiologists and ultrasonographers). Figures 2b and 3 might be combined to show the actual specimen alongside the diagram of what it shows, in which case 2b should be redrawn to follow the shape of the resection specimen.

In the Discussion, paragraph 2 might best go into the Introduction. I am not sure of the relevance of intra-thyroidal metastases, particularly from the kidney in this section, possibly a mention in the Introduction alongside the discussion of extrathyroidal extension would be more appropriate. The paragraph about targeted therapy (lines 189-195) is not really relevant and might be omitted. In the final paragraph (lines 196-205), I am not sure of the relevance of vascular invasion on a microscopic basis as involvement of microscopic vessels is not really related to invasion of large vessels (except that cancers with vascular invasion tend to be more aggressive). 

Author Response

This is an interesting case and an interesting topic generally. However, the logical flow of your presentation and particularly the discussion is lost because of your problems with writing in English. Personally, I have great respect for anyone writing in a language not their own and I do appreciate the difficulties. Sometimes it is the way a sentence is phrased, and sometimes it is single words (for example, polars, line 88; outcome, line 126; muti parametric, line 164).

This point has been corrected in the manuscript

The thrust of the case is that this is a ”tricky question” (title) or “diagnostic trap” (line 164) for the thyroid surgeon. Surely it is more a challenge for diagnosticians (radiologists and ultrasonographers mostly) as part of the assessment of any neck or thyroid lump, particularly when there is any evidence of extrathyroidal extension?

This point has been corrected in the manuscript

It illustrates that internal jugular vein tumor thrombus is an extremely rare condition in thyroid carcinoma that can be a pitfall that surgeons should not overlook in their preoperative checklist in order to anticipate surgical difficulties on the vascular level. It is a diagnostic challenge for US operators who should search it in all patients with thyroid carcinoma.

It is also better not to refer to a thyroid surgeon as male (“his” in line 151) – “the preoperative checklist” would avoid this.

This point has been corrected in the manuscript

The Introduction just repeats some of the Abstract, it would be better not to do this but to set the scene by talking about the presence of extrathyroidal extension (more usually into oesophagus or trachea when a major structure is affected) and how common (or rare) this is, and that it only occurs in a small proportion of patients with differentiated thyroid cancer (i.e. papillary / follicular) and a higher proportion of those with medullary or anaplastic cancer. That might be the point to introduce the idea that venous thrombosis can be the direct effect of tumour invasion but might alternatively relate to a hypercoagulable state. Introducing the idea of propagation of thrombus down into the SVC or cranially would be good here so that the reader can understand the risk of unrecognised venous thrombosis. It would be good also to describe here (with any appropriate references) what the consequences of unrecognised neck vein thrombosis actually are. It is less uncommon to find IJV/EJV thrombosis in head and neck cancers as an incidental finding, and in most cases, there are no consequences. The risk of pulmonary embolism (as compared to leg vein thrombosis) seems very low.

Introduction and discussion have been corrected. Appropriate references have been added in the introduction.

The actual case is well described though hard to follow in places. The use of the word “bifocal” throughout the paper is excessive. I am not sure the risks are greater from tumour invasion at two sites compared to one.

The word “bifocal” has been only cited in description of the case report.

Figure 1 (the image of  the PET/CT scan) is too small, and there should be a legend to explain what the blue circle on the image is showing. The ultrasound images in Figure 2a are less helpful (except to radiologists and ultrasonographers). Figures 2b and 3 might be combined to show the actual specimen alongside the diagram of what it shows, in which case 2b should be redrawn to follow the shape of the resection specimen.

This point has been corrected in the manuscript

In the Discussion, paragraph 2 might best go into the Introduction.

This point has been corrected in the manuscript

I am not sure of the relevance of intra-thyroidal metastases, particularly from the kidney in this section, possibly a mention in the Introduction alongside the discussion of extrathyroidal extension would be more appropriate. The paragraph about targeted therapy (lines 189-195) is not really relevant and might be omitted.

this paragraph has been removed

In the final paragraph (lines 196-205), I am not sure of the relevance of vascular invasion on a microscopic basis as involvement of microscopic vessels is not really related to invasion of large vessels (except that cancers with vascular invasion tend to be more aggressive). 

Reviewer 3 Report

This is a case report aimed at elaborating on the clinical significance of internal jugular vein thrombosis in thyroid carcinoma.

The abstract is adequate, and has listed rationale and setting details, alongside the most important findings. It needs to be reworded to present an introduction section, and a case report section. Abbreviations such as US need to be spelled out at first mention in the text. Close follow-up is a vague term. The surgeon also might be female, and third person style is advised.

Keywords need to be rewritten, since numbers are unnecessary, and keyword is written twice.

A through language check is needed throughout the manuscript.

The objectives of the study are presented clearly and the introduction section communicates the need for investigating the possible significance of internal jugular vein thrombosis, but the entire section does not list a single reference.

The manuscript sections should adhere to the CARE guidelines, available on the Care website (https://www.care-statement.org/) alongside a CARE checklist (Introduction, Case Report, Discussion and Conclusion). Informed consent and IRB approval or waiver of thereof should be stated in the Case report section.

The discussion is very comprehensive, detailed and interesting. I would support publication pending language revision, abstract rewording and adhering to CARE guidelines for content organization.

Round 2

Reviewer 2 Report

This paper does flow better after your revisions. However, there are a number of areas where this still needs some rewording.

In the Introduction (line34), it is incorrect to say that follicular carcinomas spread more frequently by lymphatic spread. This is true of papillary carcinomas but not follicular.

In line 60, I am not clear what you mean by epiduritis - perhaps “without dural involvement or compression”?

In line 67: Calcitonin looks like an extra word?

In the Case Report (line 95-99) you do not specifically say that this is a follicular carcinoma. This needs to be included. Likewise in Figure 4, it would be appropriate to include the word “follicular” in the figure legend.

References: there are 6 unnumbered references. These need to be number and referenced in the text.

In line 101: I am unclear what 33N- means. Perhaps this could be omitted?

Most of the problems are essentially linguistic. I would suggest the following changes:

Line 60 + 158: PET scanning in place of scanner

Line 61: rib in place of coast

Line 62: in, rather than of the right thyroid lobe

Line 78 + 123: diagram rather than scheme

Line 80-81: identified rather than concluded to

Line 85: Anticoagulant rather than anticoagulation

Line 88: inferiorly and superiorly rather than up and down sides

Line 89: excised rather than cut off

Line 89-90: A central compartment and right selective neck dissection (levels 2-4) was performed rather than the existing sentence

Lines 169: look for rather than take into account

Line 173: delete side (this is implied in location)

Lines 174-5: the meaning of this sentence is unclear

Line 182: “possible enlarged”. Do you mean extended (in the sense of more extensive surgery)?

Lines 184-5: I am not sure this sentence really helps in that these treatments are not specific to the tumour thrombi in the previous sentence

Line 200: you say that patients with intrathyroidal vascular invasion were more likely to develop distant recurrence. This is correct, but you need to make it clearer that for all types of thyroid cancer, the risk of locoregional recurrence and distant metastasis is greater when there is extrathyroidal extension (especially when there is clear evidence of invasion of surrounding structures). I think this is a more significant association than the presence of intrathyroidal vascular invasion.

Line 206: should report on patency of the large vessels in the neck and check for the presence of tumour thrombi.

Author Response

This paper does flow better after your revisions. However, there are a number of areas where this still needs some rewording.

In the Introduction (line34), it is incorrect to say that follicular carcinomas spread more frequently by lymphatic spread. This is true of papillary carcinomas but not follicular.

This point has been clearly corrected in the manuscript.

In line 60, I am not clear what you mean by epiduritis - perhaps “without dural involvement or compression”?

This point has been clearly corrected in the manuscript.

In line 67: Calcitonin looks like an extra word?

This point has been clearly corrected in the manuscript.

In the Case Report (line 95-99) you do not specifically say that this is a follicular carcinoma. This needs to be included. Likewise in Figure 4, it would be appropriate to include the word “follicular” in the figure legend.

This point has been clearly corrected in the manuscript.

References: there are 6 unnumbered references. These need to be number and referenced in the text.

I can't find the 6 unnumbered references you mention. All 22 references are numbered and referenced in this manuscript.

In line 101: I am unclear what 33N- means. Perhaps this could be omitted?

This point has been clearly corrected in the manuscript.

Most of the problems are essentially linguistic. I would suggest the following changes:

Line 60 + 158: PET scanning in place of scanner

This point has been clearly corrected in the manuscript.

Line 61: rib in place of coast

This point has been clearly corrected in the manuscript.

Line 62: in, rather than of the right thyroid lobe

This point has been clearly corrected in the manuscript.

Line 78 + 123: diagram rather than scheme

This point has been clearly corrected in the manuscript.

Line 80-81: identified rather than concluded to

This point has been clearly corrected in the manuscript.

Line 85: Anticoagulant rather than anticoagulation

This point has been clearly corrected in the manuscript.

Line 88: inferiorly and superiorly rather than up and down sides

This point has been clearly corrected in the manuscript.

Line 89: excised rather than cut off

This point has been clearly corrected in the manuscript.

Line 89-90: A central compartment and right selective neck dissection (levels 2-4) was performed rather than the existing sentence

This point has been clearly corrected in the manuscript.

Lines 169: look for rather than take into account

This point has been clearly corrected in the manuscript.

Line 173: delete side (this is implied in location)

This point has been clearly corrected in the manuscript.

Lines 174-5: the meaning of this sentence is unclear

This sentence was corrected: “Direct intraluminal spread" is the primary tumor invasion from the medial thyroid vein into the IJV lumen without invading the IJV wall.

Line 182: “possible enlarged”. Do you mean extended (in the sense of more extensive surgery)?

This point has been clearly corrected in the manuscript.

Lines 184-5: I am not sure this sentence really helps in that these treatments are not specific to the tumour thrombi in the previous sentence

This point has been clearly corrected in the manuscript.

Line 200: you say that patients with intrathyroidal vascular invasion were more likely to develop distant recurrence. This is correct, but you need to make it clearer that for all types of thyroid cancer, the risk of locoregional recurrence and distant metastasis is greater when there is extrathyroidal extension (especially when there is clear evidence of invasion of surrounding structures). I think this is a more significant association than the presence of intrathyroidal vascular invasion.

This point has been clearly corrected in the manuscript.

Line 206: should report on patency of the large vessels in the neck and check for the presence of tumour thrombi.

This point has been clearly corrected in the manuscript.
